# Transport of volcanic aerosol from the Raikoke eruption in 2019 through the Northern Hemisphere

Zhen Yang<sup>1,2,3</sup>, Bärbel Vogel<sup>1,3</sup>, Felix Plöger<sup>1,3</sup>, Zhixuan Bai<sup>2</sup>, Dan Li<sup>4,2</sup>, Sabine Griessbach<sup>5,3</sup>, Lars Hoffmann<sup>5,3</sup>, Frank G. Wienhold<sup>6</sup>, Elizabeth Asher<sup>7,8</sup>, Alexandre A. Baron<sup>7,9</sup>, Katie R. Smith<sup>7,9</sup>, Troy Thornberry<sup>9</sup>, Jianchun Bian<sup>2,10,11</sup>, and Michaela I. Hegglin<sup>1,3,12,13</sup>

Correspondence: Zhen Yang (yangzhen212121@gmail.com) and Bärbel Vogel (b.vogel@fz-juelich.de)

Abstract. Volcanic injections into the upper troposphere–lower stratosphere (UTLS) affect climate by altering Earth's radiation budget and atmospheric chemistry. However, the pathways by which mid-latitude eruptions spread globally remain poorly understood. We combine nighttime Compact Optical Backscatter Aerosol Detector (COBALD) profiles over Lhasa with ERA5-driven Chemical Lagrangian Model of the Stratosphere (CLaMS) backward trajectories and global three-dimensional SO<sub>2</sub>-based tracer simulations. With this integrated framework, we track the Raikoke plume (21–22 June 2019; VEI 4) as it evolved within the mature Asian Summer Monsoon Anticyclone (ASMA). Balloon-borne measurements capture the plume's arrival, vertical spreading, and dilution by ASMA-interior air. Trajectories reveal two principal pathways from distinct Raikoke plumes: (i) an upper-level branch within the summertime stratospheric easterly flow (~ 460–490 K) carrying the trailing filament of the vorticized volcanic plume (VVP), and (ii) a lower-level branch within the subtropical westerly jet (~ 390–430 K) carrying the main plume. Although the ASMA can act as a transport barrier at certain potential-temperature levels, it admits in-mixing along jet-aligned filaments and redistributes aerosols internally. SO<sub>2</sub>-based tracer simulations are sensitive to how parameterized small-scale mixing is represented in CLaMS, underscoring the need to adjust subgrid-scale mixing parameterizations when model resolution changes (here, from ERA-Interim to ERA5 reanalyses). Independent Portable Optical Particle Spectrometer (POPS) profiles over Boulder (USA) confirm the plume's timing and altitude, providing out-of-region validation. Sensitivity to injection level indicates an additional ~ 4–5 km of uplift from aerosol-radiative lofting.

<sup>&</sup>lt;sup>1</sup>Institute of Climate and Energy Systems: Stratosphere (ICE-4), Forschungszentrum Jülich, Jülich, Germany

<sup>&</sup>lt;sup>2</sup>Laboratory of Middle Atmosphere and Global Environment Observation (LAGEO), Institute of Atmospheric Physics, Chinese Academy of Sciences, Beijing, China

<sup>&</sup>lt;sup>3</sup>Centre for Advanced Simulation and Analytics (CASA), Forschungszentrum Jülich, Jülich, Germany

<sup>&</sup>lt;sup>4</sup>State Key Laboratory of Atmospheric Environment and Extreme Meteorology, Institute of Atmospheric Physics, Chinese Academy of Sciences, Beijing, China

<sup>&</sup>lt;sup>5</sup>Jülich Supercomputing Centre, Forschungszentrum Jülich, Jülich, Germany

<sup>&</sup>lt;sup>6</sup>Institute for Atmospheric and Climate Science (IAC), ETH Zurich, Zurich, Switzerland

<sup>&</sup>lt;sup>7</sup>Cooperative Institute for Research in Environmental Sciences (CIRES), University of Colorado Boulder, Boulder, CO, USA

<sup>&</sup>lt;sup>8</sup>NOAA Global Monitoring Laboratory, Boulder, CO, USA

<sup>&</sup>lt;sup>9</sup>NOAA Chemical Sciences Laboratory, Boulder, CO, USA

<sup>&</sup>lt;sup>10</sup>College of Earth and Planetary Sciences, University of Chinese Academy of Sciences, Beijing, China

<sup>&</sup>lt;sup>11</sup>College of Atmospheric Sciences, Lanzhou University, Lanzhou, China

<sup>&</sup>lt;sup>12</sup>Institute for Atmospheric and Environmental Research, University of Wuppertal, Wuppertal, Germany

<sup>&</sup>lt;sup>13</sup>Department of Meteorology, University of Reading, Reading, UK

#### 1 Introduction

The impact of volcanic eruptions on climate has been a subject of widespread concern (McCormick et al., 1995; Thompson and Solomon, 2009; Solomon et al., 2011; Bourassa et al., 2012). Large eruptions can inject significant amounts of ash, water vapor, and sulfur dioxide ( $SO_2$ ) into the upper troposphere–lower stratosphere (UTLS). While ash typically settles out within weeks,  $SO_2$  is oxidized into sulfate aerosols. These aerosols can persist in the stratosphere for months to years and be transported globally by atmospheric circulation, leading to significant climate impacts. Sulfate aerosols efficiently scatter incoming solar radiation back to space, producing a global net cooling in the lower troposphere (Minnis et al., 1993; Tabazadeh et al., 2002; Thompson and Solomon, 2009; Solomon et al., 2011). For instance, the 1991 eruption of Mount Pinatubo caused a global surface temperature decrease of approximately  $0.5^{\circ}$ C over the subsequent two years (McCormick et al., 1995). Reconstructions based on satellite observations for 1979–2015 indicate that clusters of moderate eruptions since 2004 have imposed a persistent negative forcing of about  $-0.08 \text{ W m}^{-2}$  (Schmidt et al., 2018). For 2014–2022, stratospheric injections from volcanic eruptions and wildfires produced a global-mean effective radiative forcing of about  $-0.18 \text{ W m}^{-2}$  (Yu et al., 2023). In addition, enhanced aerosol surface-area density promotes heterogeneous chemistry on polar stratospheric clouds, thereby accelerating ozone depletion (Hofmann and Solomon, 1989; Portmann et al., 1996; Solomon, 1999; Zuev et al., 2015; Solomon et al., 2016).

Several key factors govern the climate impact of volcanic aerosols from an eruption: (1) eruption magnitude; (2) injection height and self-lofting; (3) eruption latitude; and (4) dynamical evolution. (1) The Volcanic Explosivity Index (VEI) serves as a proxy for eruption intensity (Newhall and Self, 1982), and events with VEI  $\geq$  4 can inject vast quantities of SO<sub>2</sub>, water vapor, and ash, causing marked climate perturbations. (2) Eruption products injected directly into the stratosphere—or volcanic plumes in the upper troposphere that self-loft via radiative heating into the lower stratosphere—can persist far longer. (3) Aerosols from tropical eruptions are transported most efficiently via the Brewer–Dobson circulation, whereas mid-latitude eruption aerosols can still reach the tropics through Rossby-wave breaking or transport by the Asian Summer Monsoon Anticyclone (ASMA) (Konopka et al., 2009; Kloss et al., 2021; Wu et al., 2023). (4) UTLS jet streams, cyclones, anticyclones, and stratospheric circulation govern dispersion patterns and dilution rates. Note: Submarine eruptions can have high VEI yet modest stratospheric SO<sub>2</sub>; for example, Hunga (2022; VEI 5–6) injected only ~0.5 Tg SO<sub>2</sub>, while strongly perturbing stratospheric H<sub>2</sub>O and aerosol microphysics (Carn et al., 2022; Zhu et al., 2022).

Among the various transport mechanisms influencing volcanic aerosol fate, the ASMA plays a particularly important role during the boreal summer. Deep convection injects pollutants into the UTLS, where the ASMA's strong anticyclonic circulation acts as a dynamical transport barrier, trapping those air masses in its circulation. Simultaneously, the barrier is permeable, and the horizontal outflow of the ASMA can transport monsoon air masses to the extratropical UTLS (Vogel et al., 2016; Yu et al., 2017). This dual role makes the ASMA a key element in understanding aerosol dispersion in the Northern Hemisphere following volcanic eruptions. Previous work on the 2011 Nabro eruption debated whether its plume reached the stratosphere directly or was lofted by monsoon ascent. Bourassa et al. (2012) proposed that the plume remained in the upper troposphere and

was subsequently transported into the stratosphere by large-scale ascent in the monsoon, whereas later studies showed evidence of direct stratospheric injection, independent of monsoon-driven lifting (e.g., Fromm et al., 2013; Vernier et al., 2013).

The mid-latitude Raikoke volcano  $(48^{\circ}N, 153^{\circ}E)$  erupted on 21–22 June 2019 (VEI 4), and its aerosol plume was advected through the ASMA during the anticyclone's mature phase, providing an ideal case to examine how the ASMA modulates volcanic plume transport. Raikoke injected  $\sim 1.5~\mathrm{Tg}$  of  $\mathrm{SO}_2$  into the lower stratosphere (Cai et al., 2022; Vernier et al., 2024). Subsequently, ash-driven radiative heating lofted parts of the volcanic plume, raising the volcanic cloud top above  $20~\mathrm{km}$  within four days (Muser et al., 2020; Gorkavyi et al., 2021). Three days after the eruption, compact anticyclonic "vorticized volcanic plumes" (VVPs) detached from the main volcanic cloud, trapping more than half of the  $\mathrm{SO}_2$  mass (Khaykin et al., 2022). The primary VVP then rose to  $\sim 27~\mathrm{km}$ , spanning  $\sim 400~\mathrm{km}$  in width but only  $\sim 1.5~\mathrm{km}$  in depth. Meanwhile, the residual main plume was diluted at lower altitudes, resulting in two distinct Raikoke plumes.

Beginning in August 2019, frequent balloon-borne measurements were conducted over Lhasa (29.66°N, 91.14°E), when the site lay inside the ASMA at UTLS levels. The resulting profiles captured two pronounced aerosol layers at distinct altitudes, corresponding to the trailing filament of the VVP and the diluted main plume. Here, we use these observations together with high-resolution, ERA5-driven CLaMS simulations to investigate the source region, transport pathways, and transit times from Raikoke to the Tibetan Plateau measurement site. We constrain the latitude–longitude extent of the tracer injection region using satellite SO<sub>2</sub> column retrievals from TROPOMI (the Tropospheric Monitoring Instrument) aboard Sentinel-5P (Theys et al., 2024). We also incorporate Portable Optical Particle Spectrometer (POPS) data from Lhasa and Boulder (USA). This study investigates how the ASMA modulates UTLS volcanic plume transport following the Raikoke eruption. Specifically, we address the following research questions: (1) What are the key transport pathways of the Raikoke plume in the vicinity of the ASMA? (2) Which processes dominate plume dilution? Additionally, we quantify the sensitivity of the model simulations to different parameters and settings (e.g., parameterized mixing intensity, injection height, injection region).

The remainder of the paper is organized as follows: Section 2 describes data and methods; Section 3 presents balloon observations from Lhasa and Boulder; Section 4 reports CLaMS transport results and sensitivity analyses; Section 5 estimates self-lofting heights; Section 6 concludes; and Section 7 presents supplementary material.

#### 2 Data and Methods

## 2.1 Balloon-borne Instruments

Volcanic aerosol profiles at Lhasa (29.66°N, 91.14°E) were obtained during the 2019 Asian Summer Monsoon (ASM) season as part of the Sounding Water Vapor, Ozone, and Particles (SWOP) campaign. The SWOP campaign was led by the Institute of Atmospheric Physics, Chinese Academy of Sciences. Balloons were equipped with an electrochemical concentration cell (ECC) ozonesonde, a cryogenic frostpoint hygrometer (CFH), a compact optical backscatter aerosol detector (COBALD), and an iMet radiosonde. ECC measurements are not analyzed in this study. A total of nine measurements occurred in July–August (one in July, eight in August), followed by monthly measurements through January 2020. On 20 August 2019, the payload also included a Portable Optical Particle Spectrometer (POPS). During July–August 2019, Lhasa lay within the interior of


Figure 1. (a) Red markers show the geographic locations of the Raikoke volcano, Lhasa, and Boulder. (b) Spatial distribution of  $SO_2$  vertical column density (DU) from TROPOMI, retrieved with an assumed plume height of  $15 \,\mathrm{km}$ , using orbits from 22:46 UTC on 24 June to 03:50 UTC on 25 June 2019.

the ASMA at UTLS levels. The site was chosen to observe the Raikoke plume as it circulated within the anticyclone and to document its arrival and vertical evolution. Additional information on SWOP field activities and on campaigns conducted in other years is available in Bian et al. (2012); Li et al. (2017, 2018, 2020); Ma et al. (2022); Yang et al. (2023).

The POPS data over Boulder were provided by the Baseline Balloon Stratospheric Aerosol Profiles (B<sup>2</sup>SAP) project; we analyze five profiles—three influenced by the Raikoke eruption (two in August 2019 and one in November 2019) and two background references on 28 June 2019 and 3 December 2019 (Todt et al., 2023).

Figure 1a shows the geographic locations of Raikoke volcano and the balloon-sounding sites at Lhasa and Boulder.

The Compact Optical Backscatter Aerosol Detector (COBALD) was developed by ETH Zurich (Brabec et al., 2012). Its lightweight and portable design makes it suitable for balloon payloads used in the study of cirrus clouds (Brabec et al., 2012; Cirisan et al., 2014; Reinares Martínez et al., 2021; Yang et al., 2023) and aerosols (Vernier et al., 2015; Brunamonti et al., 2018; Hanumanthu et al., 2020). COBALD uses two high-power LEDs emitting at  $455 \,\mathrm{nm}$  (blue) and  $940 \,\mathrm{nm}$  (near-infrared).

It measures backscattered light from air molecules and particles, including aerosols and cloud particles (e.g. ice crystals). The backscatter ratio (BSR) is defined as

BSR = 
$$\frac{\beta_{\text{air}} + \beta_{\text{particles}}}{\beta_{\text{air}}}$$
, (1)

where  $\beta_{air}$  is the backscatter coefficient of air molecules and  $\beta_{particles}$  is the backscatter coefficient of aerosols or cloud particles. Here, BSR<sub>455</sub> represents the backscatter ratio at  $455 \, \mathrm{nm}$ . The instrument's field of view is  $\pm 6^{\circ}$ , and its detection range spans  $0.5\text{--}10 \, \mathrm{m}$ . Because sunlight severely interferes with the measurements, COBALD is only operated at night. The maximum BSR uncertainty is  $1.3 \, \%$  at  $940 \, \mathrm{nm}$  and  $0.2 \, \%$  at  $455 \, \mathrm{nm}$  at ground level. At  $10 \, \mathrm{km}$  altitude, uncertainties increase to  $5 \, \%$  and  $1 \, \%$  at  $940 \, \mathrm{nm}$  and  $455 \, \mathrm{nm}$ , respectively (Vernier et al., 2015).

The Cryogenic Frostpoint Hygrometer (CFH) is a high-precision instrument for measuring water vapor based on the chilled mirror principle. It controls the mirror temperature to maintain a stable frost- or dew-point. Combined with temperature and pressure data from the iMet radiosonde, the relative humidity over ice  $(RH_{ice})$  is calculated using the empirical equation from Murphy and Koop (2005):

$$RH_{ice} = \frac{e_{ice}(T_{mirror})}{e_{ice}(T_{environment})},$$
 (2)

where  $T_{\text{mirror}}$  is the measured frost-point temperature and  $T_{\text{environment}}$  is the ambient air temperature. The measurement uncertainty is approximately 2% in the lower troposphere and increases to 5% near the tropical tropopause (Vömel et al., 2016).

The Portable Optical Particle Spectrometer (POPS) is a lightweight instrument for measuring aerosol number density and size distribution (Gao et al., 2016). POPS employs a 405 nm laser to detect light scattered by individual particles, with scattering intensity related to particle size. It measures particles from  $140 \, \mathrm{nm}$  to  $2,500 \, \mathrm{nm}$  and reports number concentrations in size bins (Todt et al., 2023). Measurement uncertainties are dominated by sizing (including sensitivity to the assumed refractive index) and flow-rate calibration (Gao et al., 2016; Mei et al., 2020).

#### 2.2 TROPOMI



TROPOMI, the satellite instrument aboard ESA's sun-synchronous Sentinel-5P platform launched on 13 October 2017, is a hyperspectral imaging spectrometer that records back-scattered radiation from the ultraviolet to the shortwave infrared (Veefkind et al., 2012). Total-column  $SO_2$  in Dobson units (1  $DU = 2.69 \times 10^{16} \,\mathrm{molec/cm^2}$ ) is retrieved using differential optical absorption spectroscopy (DOAS) applied to three wavelength windows (312–326 nm, 325–335 nm, 360–390 nm) (Theys et al., 2017, 2024). The publicly available Level 2 product provides four vertical columns: the surface–to–top-of-atmosphere column and three columns assuming an  $SO_2$  plume centered at 1, 7, or 15 km altitude. Following previous Raikoke studies (Muser et al., 2020; de Leeuw et al., 2021; Cai et al., 2022), we adopt the 15 km retrieval, which most closely matches the eruption's mean injection height; values below the instrument's 0.3 DU detection limit are excluded (Theys et al., 2024). Four consecutive TROPOMI overpasses of the  $SO_2$  vertical column density at 15 km altitude captured the entire Raikoke  $SO_2$  cloud between 22:46 UTC on 24 June and 03:50 UTC on 25 June 2019 (Fig. 1b).

#### 2.3 CLaMS





The Chemical Lagrangian Model of the Stratosphere (CLaMS) is a chemistry–transport model that calculates the three-dimensional motion of air parcels, which in turn represent the model grid (McKenna et al., 2002; Pommrich et al., 2014). CLaMS is applied here to investigate the transport of the volcanic plume following the Raikoke eruption. In the vertical direction it employs an isentropic coordinate aligning layers with constant potential temperature θ, making the model well suited for stratospheric processes. The CLaMS vertical coordinate—the hybrid coordinate ζ—follows orography near the surface and transitions smoothly into θ once  $\sigma = p/p_{\rm surf}$  reaches 0.3 (usually about 300 hPa) (Pommrich et al., 2014). All CLaMS applications in this study—both backward trajectories and three-dimensional simulations with SO<sub>2</sub>-based tracers—are driven by ERA5 reanalysis: the main runs use the native hourly fields on the 0.3° × 0.3° grid with 137 vertical levels up to 80 km (Hersbach et al., 2020), whereas the sensitivity experiment in Appendix Figure A2 repeats the simulation with ERA5 data down-sampled to 1° × 1° and every 6 h. Compared to ERA-Interim, ERA5 improved the Lagrangian transport representation in the troposphere and cross-tropopause exchange (Hoffmann et al., 2019; Li et al., 2020), as well as within the Asian monsoon region (Clemens et al., 2024; Vogel et al., 2024).

Diabatic backward trajectories are initialized every second along the balloon's vertical ascent profile, using the in-situ measurements of temperature, pressure, time, longitude, and latitude to define the start positions. Vertical velocities were computed from ERA5 wind fields and total diabatic heating rates, including clear-sky and cloud radiation, latent heat release, and turbulent and diffusive transport (Ploeger et al., 2021). This approach captures uplifts by resolved convection as represented in ERA5 (Li et al., 2020; Clemens et al., 2024; Vogel et al., 2024). To identify the volcanic influence, only those backward trajectories that passed through the Raikoke eruption region (Fig. 1b) during 22:46 UTC on 24 June and 03:50 UTC on 25 June 2019 were retained.

The global three-dimensional CLaMS transport simulations include the CLaMS-specific parameterization of small-scale, unresolved mixing processes induced by shear-driven stirring in the large-scale flow (Konopka et al., 2004). Based on the TROPOMI  $15\,\mathrm{km}$  SO<sub>2</sub> product shown in Fig. 1b, we used the Raikoke plume's overall latitude-longitude extent (four overpasses between  $22:46\,\mathrm{UTC}$  on  $24\,\mathrm{June}$  and  $03:50\,\mathrm{UTC}$  on  $25\,\mathrm{June}$   $2019;\,\mathrm{values} \geq 0.3\,\mathrm{DU}$ ) as the tracer injection mask; only the spatial extent, not the column amplitude, was used. At  $00:00\,\mathrm{UTC}$  on  $25\,\mathrm{June}$  2019, parcels inside this mask within prescribed potential-temperature ranges  $(380-400,\,400-420,\,420-440\,\mathrm{K})$  were initialized to 1 inside the mask  $(0\,\mathrm{outside})$  for the SO<sub>2</sub>-based tracer and then advected and mixed globally, such that mixing yields fractional values between 0 and 1.

To assess how different mixing intensities influence the reconstruction of volcanic plume transport processes, two simulations were conducted: (i) a control simulation with mixing every 24h and a critical Lyapunov exponent ( $\lambda_c$ ) of 1.5; (ii) a modified simulation with mixing every 6h and  $\lambda_c=3.5$ . Primarily, we focus on the modified simulation results, since they provide better agreement with observations. Sensitivity to parameterized mixing intensity is discussed in Sect. 4.3 (modelling sensitivities). Together with the backward-trajectory analysis, these simulations provide a comprehensive view of the plume's large-scale dispersion.






#### 3 Measurement Results

The COBALD–CFH tandem provides a practical method for identifying cirrus clouds and aerosol layers (Brabec et al., 2012; Cirisan et al., 2014; Reinares Martínez et al., 2021; Yang et al., 2023). Cirrus clouds are identified using the criteria BSR $_{455}$  > 1.2 and RH $_{ice}$  > 70% (Vernier et al., 2015; Brunamonti et al., 2018; Hanumanthu et al., 2020; Yang et al., 2023). These clouds are shown as gray-shaded regions in Fig. 2. On 30 July 2019, RH $_{ice}$  reached  $\sim$  90% at 423 K (70 hPa), but no corresponding enhancement in BSR $_{455}$  was observed; thus, this layer is not classified as cirrus by our joint criterion and likely represents a moist, ice-free layer. When cirrus clouds and aerosols coexist, it becomes difficult to isolate the aerosol signal because cirrus BSR $_{455}$  values are significantly higher than those of aerosols. Regions showing enhanced BSR $_{455}$ —most likely due to Raikoke aerosols—are empirically highlighted in orange in Fig. 2. On 1 August 2019, 40 days after the Raikoke eruption, a potential enhancement appeared at 479–492 K (54–51 hPa). On 3 August, two distinct BSR $_{455}$  peaks were observed: the upper peak at 459–472 K (59–56 hPa) reached the maximum value recorded in Lhasa ( $\sim$  1.8), and the lower peak at 410–429 K (77–70 hPa) had a somewhat weaker magnitude ( $\sim$  1.4). On 6 August, another sharp peak appeared at 404–418 K (82–74 hPa). After 8 August, the peaks became less sharp and more vertically extended, which persisted until 24 November 2019. By 4 January 2020, no volcanic aerosol signal was evident.

The median BSR $_{455}$  values in 2019, associated with the Raikoke aerosol, peaked at  $\sim 1.24$ —higher than the ATAL peak in 2013 ( $\sim 1.10$ ; Fig. 3a). Typical ATAL profiles are largely confined to 360–400 K (core near 370–390 K), with occasional extensions up to 420–440 K depending on region and year (Vernier et al., 2015, 2018; Appel et al., 2022). The 2019 median peak occurs near 417 K, about 33 K above the ATAL peak (384 K), highlighting a vertical signature that differs markedly from a typical ATAL profile.

All POPS particle number densities are converted to standard temperature and pressure (STP; 1013 hPa, 273.15 K) to remove the pressure-driven decline with altitude (Fig. 3b). At Boulder, no clear volcanic aerosol signal was present on 28 June, and by 3 December the profile shows no obvious volcanic aerosol peak (Fig. 3b). The particle number density at Lhasa on 20 August was lower than that at Boulder on 7 August and 27 August. Section 4 presents backward-trajectory calculations and SO<sub>2</sub>-based tracer simulations with CLaMS to verify the origin and transport pathways of the volcanic plume reaching Lhasa within the ASMA.

**Figure 2.** COBALD (blue lines) and CFH (purple lines) measurements in Lhasa from 30 July 2019 to 4 January 2020. Dashed black lines: temperature °C. Orange shading: Raikoke eruption impact. Gray shading: presence of cirrus clouds.

Figure 3. (a) COBALD measurements in Lhasa. The black line represents the median  $BSR_{455}$  affected by the volcanic eruption in 2019. The red line shows the median  $BSR_{455}$  during the ATAL in 2013. The blue line corresponds to the  $BSR_{455}$  on 30 July 2019, which was not affected by the Raikoke eruption. The thin gray lines indicate individual profiles influenced by the volcanic event. (b) POPS measurements in Boulder and Lhasa.

## 4 Results of CLaMS transport simulations

## 4.1 Backward-Trajectory Calculations



To verify that the enhanced aerosol layer observed over Lhasa originated from the Raikoke eruption, we performed backward-trajectory analyses based on in-situ balloon-borne measurements over Lhasa, driven by high-resolution ERA5 data (Fig. 4). Backward trajectories were launched every second along the balloon's vertical ascent in the potential temperature range where an enhanced BSR<sub>455</sub> was observed in Fig. 2, and traced back to the eruption period. The start positions of the trajectories were calculated using iMet radiosonde measurements.

The enhanced BSR<sub>455</sub> was first detected over Lhasa on 1 August 2019, 40 days after the Raikoke eruption, with pronounced peaks observed on 3 August and 6 August (Fig. 2). On 1 August and for the upper layer on 3 August, backward trajectories indicate that the air parcels were transported approximately along isentropic surfaces (Fig. 4). By contrast, the lower layer on 3 August includes trajectories lofted from  $\sim 370\,\mathrm{K}$  to  $\sim 410\,\mathrm{K}$ . In the layer on 1 August and in the upper layer on 3 August, the transport pathways differed significantly from other layers, as evidenced by changes in longitude (see Fig. 5 for trajectory details). This suggests that the enhanced BSR<sub>455</sub> resulted from volcanic aerosols originating from two distinct branches of the volcanic plume at different altitudes, each following a different transport pathway. From 8–20 August, mixing from lower potential-temperature levels began to occur. This coincided with a decrease in BSR<sub>455</sub> values in the in-situ measurements,



**Figure 4.** Backward trajectories from Lhasa balloon observations to the Raikoke eruption (21–22 June 2019). Only trajectories within the enhanced BSR<sub>455</sub> potential-temperature range (orange shading in Fig. 2) are shown. Colors indicate trajectory longitude.

as air with relatively low aerosol concentrations dilutes the enhanced signals. Over the following three months, as the ASMA weakened seasonally, air from lower potential-temperature levels increasingly influenced the Lhasa profiles. During this period, volcanic signatures in the profiles no longer exhibited extreme  $BSR_{455}$  values, indicating that the enhanced volcanic aerosol layer was mixed with relatively aerosol-poor air from the lower troposphere.

To illustrate air masses influenced by the Raikoke eruption, we filtered backward trajectories to include only those passing through the eruption region (Fig. 1b) between 22:46 UTC on 24 June 2019 and 03:50 UTC on 25 June 2019. We then examined several representative pathways to show how the volcanic plume entered the ASMA interior (Fig. 5). In Fig. 5, the reported fraction is the percentage of the total backward trajectories in Fig. 4 that pass through the eruption-region mask during that window. Because this filtering criterion is highly selective, only a small fraction of trajectories remain, and the fractions can be regarded as conservative estimates.

For the calculation on 3 August 2019, the trajectories are divided into two distinct branches. One branch originates from 410–429 K and corresponds to the main volcanic aerosol plume carried by the subtropical westerly jet. The other branch originates from 459–472 K. Satellite tracking shows that the primary VVP was entrained in summertime easterlies around


**Figure 5.** Map showing representative backward trajectories originating in the volcanic eruption region, illustrating the main transport pathways from the eruption site to the Tibetan Plateau, with each trajectory shaded according to potential temperature.

20–25 July, circled the globe three times, and passed south of the Tibetan Plateau on 31 July. This pathway aligns with the backward trajectories in Fig. 5. Furthermore, the potential temperature measured by the satellite during its ASMA transit also closely matched the altitudes of enhanced BSR<sub>455</sub> (Gorkavyi et al., 2021; Khaykin et al., 2022). Meanwhile, lidar measurements in Wuhan (central China) on 30 July reveal that the VVP's core backscatter coefficient is far greater than that of the main aerosol plume (Jing et al., 2023; He et al., 2024), implying that our measurements in this branch captured only the trailing filament of the VVP. Although both sets of air parcels correspond to the same balloon-borne observation profile, they represent entirely different transport pathways—originating from two distinct regions of the Raikoke plume at different altitudes and driven by different processes.

From 6 to 20 August 2019, the overall transport pattern remained similar to that on 6 August—corresponding to the main volcanic aerosol plume primarily driven by the subtropical westerly jet. However, after entering the ASMA, some air parcels took different paths: they were advected clockwise within the ASMA's anticyclonic circulation. This clockwise advection diluted the aerosol concentration, contributing to the observed decrease in BSR<sub>455</sub>.

# 4.2 Global three-dimensional CLaMS simulations with $SO_2$ -based tracers

In global three-dimensional CLaMS simulations,  $SO_2$ -based tracers were released within the defined region (Fig. 1b) at potential-temperature levels of  $400-420\,\mathrm{K}$  at 00:00 UTC on 25 June 2019 to simulate the dispersal of the volcanic plume.

The sensitivity to injection height is discussed in Section 5 (estimation of self-lofting heights). Figure 6 shows global maps 225 with fractions of the SO<sub>2</sub>-based tracers falling within the 400-420 K layer for each measurement day from 30 July to 20 August, using the modified simulation (mixing every 6 h). Here, the fraction denotes the proportion of initialized tracer parcels that remain within the specified potential-temperature layer after advection and mixing. At initialization, the tracer is set to 1 inside the eruption-region mask and 0 elsewhere; subsequent mixing yields values between 0 and 1. The ASMA is indicated by the Montgomery streamfunction (Santee et al., 2017) at the  $378 \times 10^3 \, \mathrm{m}^2 \, \mathrm{s}^{-2}$  contour (layer mean over 400– $420 \, \mathrm{K}$ ) as an estimate 230 of its edge. The 11 PVU potential vorticity contour (layer mean over 400–420 K) indicates the dynamical situation, showing in addition outflow of the ASMA (filaments) and in-mixing of PV-rich air from high latitudes. The strong transport barrier of the ASMA observed within  $\sim 370-390\,\mathrm{K}$  is much weaker above  $400\,\mathrm{K}$  (Ploeger et al., 2015). The results shown in Fig. 6 illustrate the overall intrusion of air masses into the ASMA and agree well with observations. From 30 July to 1 August, part of the 235 SO<sub>2</sub>-based tracers entered the ASMA from its eastern side without reaching Lhasa. This is consistent with the absence of clear aerosol signals in the lower layers of the balloon-borne measurement data. On 3 August and 6 August, tracers arrived above Lhasa. Without a significant effect of upwelling from below at that time, the tracer signals remained sharp and concentrated. From 8 to 20 August, most of the ASMA contained mixed air masses. However, tracer fractions inside the ASMA remained lower than outside, reflecting the anticyclonic transport barrier.

## 240 4.3 Modelling sensitivities



To assess the sensitivity of plume dispersion to model settings, such as parameterized mixing intensity, tracer release regions, and release altitudes, we performed a series of sensitivity experiments. In previous work with CLaMS driven by  $1^{\circ} \times 1^{\circ}$ resolution and 6h temporal resolution reanalysis data (e.g., ERA5, ERA-Interim, as applied in Kloss et al. (2021)), the intensity of parameterized small-scale mixing was controlled by choosing a Lyapunov exponent of 1.5 and a mixing frequency of 24 h. Here, we performed simulations with CLaMS driven by high-resolution ERA5 data  $(0.3^{\circ} \times 0.3^{\circ} \text{ resolution}, 1 \text{ h})$  and found best agreement with observations by enhancing mixing intensity in the simulation slightly by choosing a Lyapunov exponent of 3.5 and a mixing frequency of 6 h. To evaluate the sensitivity of our results with respect to small-scale mixing intensity, Figure A1 in the Appendix shows the SO<sub>2</sub>-based tracer fractions from the control simulation (Lyapunov = 1.5; mixing = 24 h) and compares them with those from the modified simulation (Lyapunov = 3.5; mixing = 6 h) in Figure 6. The control simulation in Figure A1 shows a similar overall intrusion pattern but lacks the distinct fractions over Lhasa on 3 August. The tracer distribution in the control simulation is more fragmented, presenting small, isolated pockets compared to the more homogeneous structures in the modified simulation. Note that, for the control simulation, changing the resolution of the ERA5 driving data has only a minor effect and does not affect our conclusions; compare Fig. A1  $(0.3^{\circ} \times 0.3^{\circ}$  resolution, 1 h) with Fig. A2  $(1^{\circ} \times 1^{\circ}$  resolution, 6 h). The SO<sub>2</sub>-based tracers from global three-dimensional CLaMS simulations are interpolated in time and space to the balloonborne vertical profiles, providing a tracer-fraction profile at each measurement time. By comparing these tracer-fraction profiles from the model simulations with different mixing intensities with the observations (Fig. 7), we assess how the chosen mixing intensities affect the accuracy of the simulated plume dispersion. To quantify model-measurement agreement for each profile, we compute (i) the Pearson correlation coefficient r between the area-normalized tracer-fraction profile and the COBALD



Figure 6. Fractions of  $SO_2$ -based tracers within the  $400-420\,\mathrm{K}$  layer from the modified simulation. The purple triangle indicates Lhasa. The ASMA edge is shown by the black contour at the  $378\times10^3\,\mathrm{m}^2\,\mathrm{s}^{-2}$  Montgomery streamfunction. The white contour marks the  $11\,\mathrm{PVU}$  potential vorticity boundary.

BSR<sub>455</sub> enhancement (defined as BSR<sub>455</sub>-1) over the main plume layer (375–450 K; 375–475 K on 30 September, 28 October, and 24 November), and (ii) the absolute peak-height difference  $|\Delta\theta|$  between the two profiles. Note that r reflects only linear association and should be interpreted as a relative indicator rather than a definitive measure of agreement. The values of r and  $|\Delta\theta|$  are annotated in Fig. 7. In Fig. 7, the SO<sub>2</sub>-based tracers are also released at 400–420 K. This setup primarily samples the diluted main plume rather than the higher-altitude trailing filament of the vorticized volcanic plume (VVP). Accordingly, the higher-altitude peak on 3 August 2019 likely originates from the VVP filament near 460–490 K and is not resolved by the 400–420 K release. Consistent with Figures 6 and A1, the modified simulation produces tracer peaks (blue lines) that closely match the balloon measurements—except on 6 August and 8 August, when the peak altitudes deviate slightly. For most dates, the modified simulation shows consistently higher r and smaller  $|\Delta\theta|$ . Although the control simulation (orange lines) occasionally aligns with observed peak altitudes, its overall agreement is weaker and it even generates unexplained extreme



values on 12 August and 24 November. Overall, the modified simulation captures the vertical structure of the  $BSR_{455}$  profile well, despite minor differences in peak altitude for certain dates.

Furthermore, we assess the sensitivity of model simulations to the plume initialization by carrying out another simulation with SO<sub>2</sub>-based model tracers initialized within a rectangular latitude–longitude mask (from 163°E to 170°W and 49°N to 62°N), following Kloss et al. (2021). Tracers were then released on multiple isentropic levels within this rectangular domain. Here, we show the rectangular-mask tracers, calculated by the modified simulation, released at potential-temperature levels of 400–420 K during 23–24 June 2019 (red lines in Fig. 7). The peak heights of the rectangular tracers (red lines) and the SO<sub>2</sub>-based tracers (blue lines) show no significant difference. Because the release domain captures the plume core, even a simple rectangular mask reproduces the observed peak altitude. Hence, for the Raikoke simulations the choice of injection region is less critical than the choice of model mixing intensity.

Over Boulder, we likewise compare the model tracer-fraction profiles on the observation dates with the POPS aerosol number concentration profiles at STP (Fig. 8). For the computation of r, we evaluate within 375–450 K for 7 August and 27 August, and within 375–475 K for 8 November 2019. On 27 August 2019, the SO<sub>2</sub>-based tracers in the modified simulation (blue) closely reproduce the observed peak. On 7 August 2019, judging by the reproduced peak shape, the SO<sub>2</sub>-based tracers from the modified simulation still agree best relative to the SO<sub>2</sub>-based tracers in the control simulation and to the rectangular-mask tracers in the modified run.

Figure 7. Comparison of tracer-fraction profiles with the BSR<sub>455</sub> profile over Lhasa. The labels SO<sub>2</sub>-based\_Control (orange) and SO<sub>2</sub>-based\_Modified (blue) denote SO<sub>2</sub>-based tracers from the control and modified simulations, respectively, while Rectangle\_Modified (red) denotes tracers from the rectangular release domain in the modified simulation. Annotation colors match the corresponding simulation profiles. Values give the Pearson correlation r and the absolute peak-height offset  $|\Delta\theta|$  (K) between the model tracer-fraction profile and the COBALD BSR<sub>455</sub> profile within the analyzed  $\theta$  range (375–450 K; 375–475 K on 30 September, 28 October, and 24 November).

Figure 8. Same as Fig. 7 but for Boulder: comparison of tracer-fraction profiles with the POPS aerosol number-concentration profile at STP. Annotations report r and  $|\Delta\theta|$  (K) relative to POPS within the analyzed  $\theta$  range (375–450 K for 7 August and 27 August, and 375–475 K for 8 November 2019).

## 285 5 Estimation of self-lofting heights

The Raikoke plume was injected primarily within 8–16 km (Kloss et al., 2021; Cai et al., 2022; Vernier et al., 2024). Radiative heating of ash and nascent sulfate then lofted parts of the plume, raising volcanic cloud tops above 20 km within four days





of the eruption (Muser et al., 2020; Gorkavyi et al., 2021). These observations indicate that Raikoke's plume experienced an intense updraft during the eruption stage. Aerosol-radiative lofting is not included in the CLaMS simulations. However, comparison of the vertical profiles of the  $SO_2$ -based tracers with in-situ measurements allows estimation of the lofting height.  $SO_2$ -based tracers were injected at  $20\,\mathrm{K}$  intervals between  $360\,\mathrm{K}$  and  $500\,\mathrm{K}$  to cover the entire possible extent of the Raikoke plume. To assess which initialization range best reproduces the observed plume, we compare tracer-fraction profiles with the  $BSR_{455}$  profiles in Figure 9.

Overall, tracers initialized within 400– $420\,\mathrm{K}$  (blue lines;  $\sim 15$ – $16.5\,\mathrm{km}$  at the Raikoke site) best match the enhanced BSR<sub>455</sub> profiles, compared with 380– $400\,\mathrm{K}$  (brown) and 420– $440\,\mathrm{K}$  (purple). For most dates, the 400– $420\,\mathrm{K}$  initialization attains the highest r and the smallest  $|\Delta\theta|$  among the three layers (see annotations in Fig. 9). In one case, for example on August 3, August 6 and August 12, tracers initialized in the 400– $420\,\mathrm{K}$  layer contribute most strongly to the peak of the BSR<sub>455</sub> profile, while those from the 380– $400\,\mathrm{K}$  and 420– $440\,\mathrm{K}$  layers also produce peaks at the same altitude but with smaller amplitudes. In another case, on 8 August, 30 September, and 28 October 2019, only the 400– $420\,\mathrm{K}$  tracers reproduce the BSR<sub>455</sub> peak at the correct altitude, whereas the peaks from the 380– $400\,\mathrm{K}$  and 420– $440\,\mathrm{K}$  tracers appear below or above the observed height. For Boulder (Fig. 10), the 400– $420\,\mathrm{K}$  initialization likewise shows the best agreement with the observations. On 7 August 2019 the 380– $400\,\mathrm{K}$  layer, and on 27 August 2019 the 420– $440\,\mathrm{K}$  layer, produce peaks that disagree strongly with the observations (r 

Figure 9. Tracer-fraction profiles from the modified simulation at different isentropic injection levels, compared with COBALD BSR<sub>455</sub> profiles. Blue lines: injections at 400– $420 \,\mathrm{K}$ ; brown lines: 380– $400 \,\mathrm{K}$ ; purple lines: 420– $440 \,\mathrm{K}$ . Annotation colors match the corresponding injection-layer curves; values are r and  $|\Delta\theta|$  (K) between the model tracer-fraction profile and the COBALD BSR<sub>455</sub> profile within the analyzed  $\theta$  range (375– $450 \,\mathrm{K}$ ; 375– $475 \,\mathrm{K}$  on 30 September, 28 October, and 24 November).

Figure 10. Same as Fig. 9 but for Boulder: tracer-fraction profiles from the modified simulation at different isentropic injection levels compared with the POPS aerosol number-concentration profile at STP. Blue:  $400-420\,\mathrm{K}$ ; brown:  $380-400\,\mathrm{K}$ ; purple:  $420-440\,\mathrm{K}$ . Annotations report r and  $|\Delta\theta|$  (K) relative to POPS within the analyzed  $\theta$  range ( $375-450\,\mathrm{K}$  for 7 August and 27 August, and  $375-475\,\mathrm{K}$  for 8 November 2019).

### 310 6 Conclusions

In this study, we combined in-situ balloon observations over Lhasa and Boulder with Lagrangian transport simulations using the CLaMS model, driven by high-resolution ERA5 data, to investigate the transport pathways and mixing processes of the

Raikoke plume in the upper troposphere–lower stratosphere (UTLS) during the mature phase of the Asian Summer Monsoon Anticyclone (ASMA). Our main conclusions are:

- Analysis of backward trajectories. 40 days after the Raikoke eruption, the enhanced BSR<sub>455</sub> was first detected over Lhasa. Two separate transport pathways carried the Raikoke plume to the Tibetan Plateau: (i) a high-altitude pathway along summertime easterly winds (~ 460–490 K) transporting the trailing filament of the vorticized volcanic plume (VVP) and (ii) a lower-level route via the subtropical westerly jet (~ 390–430 K) transporting the main volcanic plume. These branches converge over Lhasa on 3 August 2019 but at different altitudes, explaining the dual peaks observed in balloon-borne backscatter profiles.
  - Dilution via mixing with relatively aerosol-poor air. During stratospheric transport, the plume is progressively diluted by background air with low aerosol concentrations. After entering the ASMA, both upwelling from lower potential-temperature levels and horizontal entrainment of surrounding air contribute to this dilution. These processes are consistent with the progressively smoothed peaks of enhanced BSR<sub>455</sub> signals observed along the balloon profiles.
- Validation of CLaMS SO<sub>2</sub>-based tracers. CLaMS modified simulations using SO<sub>2</sub>-based tracers successfully capture transport into the ASMA and reconstruct the vertical structure observed over Lhasa. Compared with the control simulation (mixing every 24 h), the modified simulation (mixing every 6 h) produces more coherent tracer distributions that closely match the enhanced BSR<sub>455</sub> peaks. By comparing SO<sub>2</sub>-based tracers released at different isentropic levels, we infer that aerosol-radiative lofting may produce an additional uplift of ~ 4–5 km.
- In summary, by combining observations and modelling, we provide further insights into how the mid-latitude Raikoke plume is transported, evolves, and spreads out over long distances, enters the ASMA, and is progressively diluted. We identified two distinct isentropic transport pathways and quantified the dilution mechanism via mixing from air masses outside the plume. Our simulations show that small-scale atmospheric mixing processes are critical for dispersing the volcanic plume along its pathway through the stratosphere, and that representing these (often subgrid-scale) processes in models is essential for reliable simulations of volcanic plume transport. In this sense, changes in model resolution require adjustment of mixing parameterizations. In particular, our findings show that the ASMA may play an important role in dispersing aerosols from mid-latitude volcanic injections throughout the global stratosphere.

Data availability. Sentinel-5P TROPOMI SO<sub>2</sub> Level-2 OFFL data were obtained from the Copernicus Data Space Ecosystem (last access: 3 June 2025); the product DOI is https://doi.org/10.5270/S5P-74eidii. ERA5 reanalysis was retrieved from the Copernicus Climate Data Store. We used the "ERA5-complete" dataset on the native grid including model levels (https://doi.org/10.24381/cds.143582cf; last access: 3 June 2025). POPS (Baseline Balloon Stratospheric Aerosol Profiles, B<sup>2</sup>SAP) 1 Hz time series were downloaded from NOAA CSL (https://csl.noaa.gov/projects/b2sap/data.html, last access: 3 July 2025); see also the dataset description by Todt et al. (2023).

# Appendix A

Figure A1. Same as Fig. 6, but for SO<sub>2</sub>-based tracer fractions from the control simulation for each measurement day, driven by ERA5 at  $0.3^{\circ} \times 0.3^{\circ}$  spatial and 1-h temporal resolution.

**Figure A2.** Same as Fig. A1, but driven by ERA5 at  $1^{\circ} \times 1^{\circ}$  spatial and 6-h temporal resolution.

Author contributions. ZY, BV, and FP jointly developed the study concept; BV and FP provided the initial idea and scientific guidance throughout; ZY performed the simulations and analyzed the data and results; BV contributed key components of the CLaMS simulation code; FP contributed the SO<sub>2</sub>-based tracer code in CLaMS; JCB, ZXB, and DL conducted the balloon-borne measurements at Lhasa; SG and LH provided the TROPOMI data; FGW provided technical support for the COBALD instrument/data; EA, AAB, KRS, and TT provided technical support for the POPS data; MIH contributed to the review and editing of the manuscript; ZY wrote the manuscript with input from all co-authors.

Competing interests. The authors declare that they have no competing interests.

Acknowledgements. This research was supported by the Deutsche Forschungsgemeinschaft (DFG grant no. VO 1276/6-1) and the National Natural Science Foundation of China (NSFC Grant 42061134012) in the frame of a joint NSFC-DFG research project as well as by the NSFC Grant 42394121.

We acknowledge the Jülich Supercomputing Centre (JSC; Research Centre Jülich, Germany) for computing time on the supercomputer JUWELS (project CLaMS-ESM) and storage resources on the meteocloud data archive. Further, we thank the European Centre for Medium-Range Weather Forecasts (ECMWF) for providing the ERA5 reanalyses.

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
