# Peer review of "Transport of volcanic aerosol from the Raikoke eruption in 2019 through the Northern Hemisphere"

_EGUsphere, 2025_

## Referee Comment (RC2)

**Review of "Transport of volcanic aerosol from the Raikoke eruption in 2019 through the Northern Hemisphere" by Zhen Yang et al. (egusphere-2025-4842)**

**General comments**

Yang et al. investigate the transport pathways of the volcanic plume emitted by the Raikoke eruption in 2019, with focus on the Asian summer monsoon anticyclone (ASMA) region. The analysis is based on a combination of aerosol backscatter measurements by COBALD sondes flown in Lhasa (China) and model simulations using the CLaMS model. Two types of simulations are presented: backward trajectories, aimed to trace back the observed aerosol plumes to the Raikoke eruption, and global tracer simulations, investigating transport in the model and its sensitivity to different parameters. Aerosol number density profiles from POPS sondes flown in Boulder (USA) are also used to evaluate the results of the tracer simulations.

The paper is well written and fits well the scope of ACP. The observational data are of high quality and nicely combined with the model simulations, including the use of TROPOMI $SO_2$ satellite retrievals to define the initial state of the eruption. The results are presented clearly and concisely, although some of the assumptions made would require in my opinion further elaboration. The figures are excellent.

I support the publication of the paper. However, I have some comments that I think should be addressed to improve and/or clarify a few aspects of the study, mainly related to the backward trajectory analysis and the use of COBALD data to define the volcanic plume, as well as the overall impact of the paper.

1. The "empirical" definition of the regions of enhanced $BSR_{455}$ attributed to the Raikoke eruption (orange shadings in Fig. 2), which are used to initialize the backward trajectories, requires some clarification. I generally agree with the identified regions, but I think the criteria used for their classification should be discussed in more detail and could be at least partly quantified.

   The main issue is the separation of the volcanic signal from aerosol features related to ATAL. There are enhanced $BSR_{455}$ features in the profiles shown in Fig. 2 that are not classified as volcanic plume, likely because they are related to ATAL, but no explanation is given on why these regions are not considered. For example, on 10-08-2019, an enhanced $BSR_{455}$ layer starts at ~140 hPa, well below the marked onset of the volcanic plume (84 hPa). Other small features can be seen on 01-08-2019 (~80-130 hPa), 06-08-2019 (~120-140 hPa), and 12-08-2019 (~120-140 hPa). If my interpretation is correct and these features are related to ATAL, I would suggest to discuss this in the text and to highlight them accordingly in Fig. 2 (e.g., by a different color shading). If not, why are these regions excluded?

   Secondly, the visual identification of the plume boundaries becomes difficult when the $BSR_{455}$ gradient is smooth, as in the last three flights (Sept-Nov). Here, I have the impression that the lower boundary of the plume does not match the onset of the $BSR_{455}$ enhancement (roughly at 100 hPa), but rather some given threshold in $BSR_{455}$. Is there a reason to define the lower boundary this way?

   One possibility to quantify these criteria could be the use of the COBALD color index (CI), defined as the ratio of the 940-to-455 nm aerosol BSR (i.e., BSR – 1). Using information from both wavelengths, the CI is a proxy of particle size that was used in several studies to separate clouds from aerosols (e.g., Vernier et

al., 2015; Brunamonti et al., 2018; Hanumanthu et al., 2020). Here, the CI might help to distinguish the volcanic plume from ATAL, assuming they have different size distributions, and to define its boundaries more accurately. Have you considered looking into this?

2. Since the Boulder profiles are not included in the backward trajectory analysis, the paper focuses almost entirely on transport of the volcanic plume in the ASMA region, rather than the whole Northern Hemisphere (as hinted by the title). The Boulder data are only used to evaluate the results of the tracer simulations, and although the agreement with the model is remarkable (Fig. 8), the discussion of these measurements remains very limited (a few lines on pages 7 and 14). At the same time, the tracer distribution map in Fig. 6 shows that the filament advected over Lhasa represents only a small fraction of the volcanic plume, while most of it stays outside of the ASMA. Therefore, I find that the general discussion is not properly balanced in this respect. I would suggest to try better integrating the Boulder data into the "storyline" of the paper, for example, by adding backward trajectories initialized from these flights into Fig. 5 (or an additional figure). This would allow to frame the entire transport pathway analysis in a more general context, and would be in my opinion a great addition to the paper. Otherwise, it should be at least pointed out more clearly that the ASMA pathway only accounts for a minor fraction of the entire volcanic plume emitted by the Raikoke eruption.

**Specific comments**

Page 1, line 15: this sentence requires more context (what does "additional" uplift refer to?).

Page 2, line 32: consider omitting the definition of VEI (not relevant to this study).

Page 2, line 44: add some references for the chemical trapping in the ASMA, e.g. Park et al. (2007), Randel et al., (2010) (see reference list below).

Page 3, line 79: which model of iMet radiosonde was used?

Page 3, line 80: I suggest adding a table to summarize date, time, location and payload of all analyzed balloon flights, and possibly to introduce a sequential numbering of the flights (e.g. F1, F2, ...).

Page 5, lines 98-100: I cannot find the source of the COBALD uncertainties given here in Vernier et al. (2015), neither the "maximum BSR uncertainties" of 1.3 % at 940 nm and 0.2 % at 455 nm at ground level, nor the 5 % at 940 nm and 1 % at 455 nm at 10 km altitude. How are these numbers obtained? Vernier et al. (2015) only estimate a 5 % uncertainty for the entire profile, due to physical constraints, and 1 % precision in the UTLS, without differentiating between the two channels. The same uncertainty of 5 % is also reported by other studies using COBALD data in the lower troposphere (Brunamonti et al., 2021) and UTLS (Reinares Martínez et al., 2021). Considering that the retrieval algorithm of COBALD BSR involves empirically-determined instrumental parameters as well as the measured temperature and pressure to calculate the molecular extinction profile, I doubt that such high accuracies can be achieved.

Page 5, lines 106-107: I would be more conservative with the CFH uncertainty. Vömel et al. (2016) state that the uncertainty "may be" as low as 2 % in the lower troposphere and 5 % at the tropical tropopause, under good operating conditions of the mirror temperature controller. The mirror temperature controller is the largest source of uncertainty in CFH measurements and oscillations around the real frostpoint can be up to ±0.5 K, corresponding to ±10 % in $H_2O$ mixing ratio at UTLS conditions (e.g., see Poltera et al., 2025). Based on Fahey et al. (2014), I think a more realistic estimate of the CFH uncertainty in the stratosphere is ±10 %, unless the performance of the mirror temperature controller is evaluated specifically for each flight.

Page 6, line 138: is temperature really needed to define the starting positions?

Page 6, line 152: add a short explanation of the physical meaning of the critical Lyapunov exponent. Does a higher $\lambda_c$ correspond to more or less mixing?

Page 7, lines 159-160: the cloud-filtering criteria used here ($BSR_{455} > 1.2$, $RH_{Ice} > 70$ %) are those derived by Yang et al. (2023), which are, to my understanding, a modified version of the criteria used in previous studies (Vernier et al., 2015; Brunamonti et al., 2018; Hanumanthu et al., 2020), without taking the color index (CI) into account. As I already mentioned, the spectral information contained in the CI is crucial to make a physically-based (rather than empirical) discrimination, since it allows to distinguish size effects (change in BSR and CI) from number density effects (change in BSR but no change in CI). Therefore, I think it would be very interesting to investigate the CI here, as this may provide a quantitative basis for a more accurate definition of the volcanic plume and its boundaries.

Page 7, line 163: if "coexist" means that an aerosol layer and a cirrus cloud overlap in altitude, then the two signals cannot be distinguished (rather than "it becomes difficult"). If they coexist in the same profile but on different altitude levels, then the visual identification may become more difficult, but the signals can still be isolated quantitatively (e.g., using the CI). Please clarify.

Page 7, line 165: I suggest "determined by visual inspection" instead of "empirically highlighted".

Page 7, lines 171-172: I presume the ATAL profile from 2013 shown in Fig. 3a is the COBALD profile from Lhasa by Vernier et al. (2015). Is this correct? Please add a citation.

Page 7, line 174: how much does 33 K potential temperature correspond in altitude (roughly)?

Page 10, line 199: quantify "extreme" $BSR_{455}$ values.

Page 10, lines 204-206: why should the filtering criterion be considered "highly selective"? Is this related to the spatial/temporal extent of the mask, or its "patchiness"? Would it help to use a more compact domain (e.g., the rectangular mask used in Section 4.3), or to extend the considered time window? And what are the source regions of the > 90 % trajectories that are not shown? This is a key point of the paper, so I think some more elaboration is required.

Page 10, line 209: which satellite? Is this statement referring to Khaykin et al. (2022)?

Page 11, lines 218-221: why are the backward trajectories of the September-November flights not included?

Page 12, line 248: I suggest adding a table to summarize the different model runs and their main characteristics (mixing parameters, injection region/height, etc.). I would also recommend introducing a more compact notation for these simulations (e.g., "SO$_2$-based_control" → CTRL$_{SO2}$), and perhaps a more meaningful name for the "modified" scenario (e.g. "mixing-enhanced", "MIX-ENH" or similar).

Page 12, lines 250-252: any idea why is the tracer distribution in the control run more fragmented?

Page 13, line 259: the quantity "BSR – 1" is usually termed "aerosol BSR" (ABSR: Cirisan et al., 2014) or particle BSR (PBSR: Reinares Martínez et al., 2021), as it represents the ratio of aerosol-to-molecular backscatter coefficient (since BSR = ($\beta_{aer}$ + $\beta_{mol}$) / $\beta_{mol}$). The term "enhancement" instead typically refers to elevated values over a given background. Please revise this definition.

Page 13, line 268: I cannot see "extreme" values in Fig. 7. Please quantify.

Page 13, figure 6: it would be nice to show the tracer distribution over the Boulder site, since these simulations are also compared with the POPS data. Consider expanding the X-axis of Fig. 6 to include the entire Northern Hemisphere, or adding an extra figure focused on the American continent.

Page 16, lines 304-309: I struggle to follow this paragraph and the argument of the "4-5 km additional lofting". If I understand correctly, this refers to the difference between the peak injection height of ~11 km derived from Cai et al. (2022), and the injection level inferred from the tracer simulation that best matches the observed profiles (400-420 K, i.e. 15-16.5 km). But how to be sure that this is due to radiative heating and not some other artifact/discrepancy between the different techniques? And what would be the time frame and the associated heating rates of the lofting? Since the plume has a large vertical extent (5-15 km), I would speculate that heating rates have a complex altitude dependency, so just comparing the peak height is not necessarily a good assumption. Unless this argument can be supported by a more detailed analysis, I think it should be presented as a speculation rather than a finding of the paper.

Page 18, lines 333-334: what are exactly the "small-scale" and "often subgrid-scale" mixing processes?

Page 18, lines 336-337: I would appreciate a few statements on the overall relevance and fate of the volcanic aerosols entrained in the ASMA vortex, beyond the fact that this "may play an important role" in dispersing the aerosols. Is there a special relevance of this filament compared to the rest of the volcanic plume, e.g., in terms of more efficient aerosol transport to the stratosphere via the ASMA dynamics (see Vogel et al., 2019), higher cold-point tropopause, etc.? If so, this would be interesting to discuss.

**Technical comments**

Page 2, line 40: the full name of the volcano is "Hunga Tonga" or "Hunga Tonga-Hunga Ha'apai".

Page 3, line 72: replace "Section 7" with "Appendix A".

Page 3, line 75: delete coordinates of Lhasa (already given).

Page 5, line 105: the quantity $e_{ice}$ (saturation vapor pressure over ice) is not defined.

Page 5, line 114: delete "13 October" (not relevant).

Page 6, line 128: "an isentropic coordinate aligning layers" check grammar.

Page 11, line 211: add "of the plume" after "potential temperature".

Page 12, lines 248-229: use the abbreviation of the Lyapunov exponent ($\lambda_c$) defined in Section 2.3.

**References (note: only papers not cited in the original manuscript are listed)**

Brunamonti, S., Martucci, G., Romanens, G., Poltera, Y., Wienhold, F. G., Hervo, M., Haefele, A., and Navas-Guzmán, F.: Validation of aerosol backscatter profiles from Raman lidar and ceilometer using balloon-borne measurements, Atmos. Chem. Phys., 21, 2267–2285, https://doi.org/10.5194/acp-21-2267-2021, 2021.

Fahey, D. W., Gao, R.-S., Möhler, O., Saathoff, H., Schiller, C., Ebert, V., Krämer, M., Peter, T., Amarouche, N., Avallone, L. M., Bauer, R., Bozóki, Z., Christensen, L. E., Davis, S. M., Durry, G., Dyroff, C., Herman, R. L., Hunsmann, S., Khaykin, S. M., Mackrodt, P., Meyer, J., Smith, J. B., Spelten, N., Troy, R. F., Vömel, H., Wagner, S., and Wienhold, F. G.: The AquaVIT-1 intercomparison of atmospheric water vapor measurement techniques, Atmos. Meas. Tech., 7, 3177–3213, https://doi.org/10.5194/amt-7-3177-2014, 2014.

Park, M., Randel, W. J., Gettelman, A., Massie, S. T., and Jiang, J. H.: Transport above the Asian summer monsoon anticyclone inferred from Aura Microwave Limb Sounder tracers, J. Geophys. Res., 112, D16309, https://doi.org/10.1029/2006JD008294, 2007.

Poltera, Y., Luo, B., Wienhold, F. G., and Peter, T.: The "Golden Points" and nonequilibrium correction of high-accuracy frost point hygrometers, EGUsphere [preprint], https://doi.org/10.5194/egusphere-2025-2003, 2025.

Randel, W. J., Park, M., Emmons, L., Kinnison, D., Bernath, P., Walker, K. A., Boone, C., and Pumphrey, H.: Asian Monsoon Transport of Pollution to the Stratosphere, Science, 328, 611– 613, https://doi.org/10.1126/science.1182274, 2010.

Vogel, B., Müller, R., Günther, G., Spang, R., Hanumanthu, S., Li, D., Riese, M., and Stiller, G. P.: Lagrangian simulations of the transport of young air masses to the top of the Asian monsoon anticyclone and into the tropical pipe, Atmos. Chem. Phys., 19, 6007–6034, https://doi.org/10.5194/acp-19-6007-2019, 2019.